# Application of Pulsed Electric Fields and High-Pressure Homogenization in Biorefinery Cascade of *C. vulgaris* Microalgae

**DOI:** 10.3390/foods11030471

**Published:** 2022-02-05

**Authors:** Daniele Carullo, Biresaw Demelash Abera, Mariarosa Scognamiglio, Francesco Donsì, Giovanna Ferrari, Gianpiero Pataro

**Affiliations:** 1Department of Industrial Engineering, University of Salerno, Via Giovanni Paolo II, 132-84084 Fisciano, Italy; danielecarullo91@gmail.com (D.C.); biresawdemelash.abera@natec.unibz.it (B.D.A.); mrscogna@unisa.it (M.S.); fdonsi@unisa.it (F.D.); gferrari@unisa.it (G.F.); 2ProdAl S.c.a.r.l., University of Salerno, Via Giovanni Paolo II, 132-84084 Fisciano, Italy

**Keywords:** *Chlorella vulgaris*, pulsed electric fields (PEF), high-pressure homogenization (HPH), biorefinery, proteins, carbohydrates, lipids

## Abstract

In this study, a cascaded cell disintegration process, based on pulsed electric fields (PEF - 20 kV/cm, 100 kJ/kg_SUSP._) and high-pressure homogenization (HPH - 150 MPa, 5 passes) was designed for the efficient and selective release of intracellular compounds (water-soluble proteins, carbohydrates, and lipids) from *C. vulgaris* suspensions during extraction in water (25 °C, 1 h) and ethyl acetate (25 °C, 3 h). Recovery yields of target compounds from cascaded treatments (PEF + HPH) were compared with those observed when applying PEF and HPH treatments individually. Particle size distribution and scanning electron microscopy analyses showed that PEF treatment alone did not induce any measurable effect on cell shape/structure, whereas HPH caused complete cell fragmentation and debris formation, with an undifferentiated release of intracellular matter. Spectra measurements demonstrated that, in comparison with HPH alone, cascaded treatments increased the selectivity of extraction and improved the yields of carbohydrates and lipids, while higher yields of water-soluble proteins were measured for HPH alone. This work, therefore, demonstrates the feasibility of sequentially applying PEF and HPH treatments in the biorefinery of microalgae, projecting a beneficial impact in terms of process economics due to the potential reduction of the energy requirements for separation/purification stages.

## 1. Introduction

In the last decades, microalgae have attracted increasing attention from the industry, because of their exploitation as an alternative source of different nutrient and non-nutrient compounds, to be used in replacement of increasingly depleted conventional sources of natural foodstuff [1,2,3,4]. Microalgae are capable of synthesizing high amounts of lipids, proteins, carbohydrates, and pigments, which could find large applications not only in the food sector but also in other markets [5,6,7,8]. For example, microalgal lipids may serve as a source for biofuels, building blocks in the chemical industry, and value-added edible oils for the functional food and health market [9,10]. Moreover, microalgal proteins, carbohydrates, and pigments may find application in the food, feed, health, and bulk chemical market, or for the production of ethanol and chemicals [11,12,13,14,15].

Because of the different properties of the intracellular molecules found in microalgae, whose mass distribution greatly depends on the considered species, as well as on applied cultivation conditions [16,17], microalgae processing can be suitably carried out through a wet-route “biorefinery” scheme, to obtain different classes of compounds upon a multi-stage downstream processing phase, involving mild and economically feasible separation steps [18,19,20,21,22].

In the biorefinery approach, the first and most crucial step after microalgae harvesting is the cell disintegration pre-treatment, through which damages to the cell wall/membrane system are induced, to reduce the mass transfer resistance in the extraction of valuable compounds from both cytoplasm and internal organelles [5,18,23], while maintaining high quality and purity of the extracts, as well as to prevent the reduction of the product value [24].

Therefore, a progressive permeabilization strategy, which can be tailored for each specific strain, has emerged as the cornerstone front of the full valorization of microalgal biomass through an efficient and sustainable biorefinery process [5,20,24,25].

Previous studies have successfully demonstrated the potential of pulsed electric fields (PEF) to gently permeabilize the envelope of different microalgae, to enable the selective recovery of specific compounds without the formation of cell debris and, hence, simplifying downstream processing [5,6,23,26,27,28,29,30,31]. For example, Postma et al. [24] showed that the electro-permeabilization effect induced by PEF significantly increased the release of small molecules, such as ions and carbohydrates, from *Chlorella vulgaris* biosuspensions, in comparison with untreated samples. However, because of the hard structure of the cell wall of *C. vulgaris*, composed of a cellulose and hemicellulose bilayer [16], the applied electrical conditions (20 kV/cm of electric field strength, and 50–100 kJ/kg_SUSP._ of specific energy input) were not sufficient to promote the extraction yield of relatively high molecular weight compounds (e.g., proteins) [24]. To avoid excessively severe processing conditions, the authors suggested using PEF as the first step of a hypothetical cascade microalgal permeabilization process, for the recovery of carbohydrates, followed by a more disruptive and efficient technology, such as bead milling [32,33], for the recovery of the remaining intracellular compounds.

In a previous work, we explored comparatively the effect of PEF and high-pressure homogenization (HPH) treatments on the permeabilization degree, morphological properties, and extractability of different compounds from *C. vulgaris* microalgae [5]. HPH was characterized by a significantly higher disruption efficiency than PEF, being ascribed to the extremely intense fluid-dynamic stresses applied [34,35], causing the instantaneous and undifferentiated release of intracellular matter. In the same work, optimal cell disruption conditions maximizing extraction yields of water-soluble proteins and carbohydrates were identified for individual PEF (20 kV/cm, 100 kJ/kg_SUSP._) and HPH (150 MPa of pressure, five passes) treatments [5].

Interestingly, the integration of PEF and HPH treatments in a microalgal “biorefinery” process represents a promising option to exploit the selectivity towards small molecular weight compounds (e.g., carbohydrates, lipids) of PEF to reduce the energy consumption in downstream processing for separation/purification purposes, and the disruption efficiency of HPH to recover, with high yields, bulky proteins.

Despite several authors having reported examples of microalgal processing for the recovery of value-added compounds, based either on PEF alone or its combination with mechanical treatments [6,20,36,37,38,39], to the best of our knowledge, this is the first attempt to develop an entire microalgal biorefinery scheme, integrating PEF and HPH technologies with different post-processing methods, to pursue the complete valorization of microalgal biomass through enhanced selectivity and yield of extraction (Figure 1).

Based on these premises, the principal aim of this work was investigating the effect of PEF and HPH technologies, applied in a cascade scheme, on the purity and recovery of valuable compounds from *C. vulgaris* microalgae during aqueous or organic extractive diffusion steps. Specifically, the impact of either single or cascaded treatments on microalgal morphological aspects, and the extraction yield of water-soluble proteins, carbohydrates, and lipids, were assessed.

## 2. Materials and Methods

### 2.1. Microalgal Strain and Cultivation

Biomass of *C. vulgaris* (CCAP 211) was kindly supplied by the Department of Civil, Chemical, and Environmental Engineering of the University of Genoa (Genova, Italy). The strain *C. vulgaris* (CCAP 211), utilized in this work, was purchased from the Culture Collection of Algae and Protozoa (Argyll, UK). Cultivation and harvesting conditions, as well as the composition of the growing medium, were already reported in our previous study [5]. Briefly, microalgae were harvested from a 5 L tubular photobioreactor (adopted light intensity ≈ 72 µmol m^−2^ s^−1^) at a concentration of 3 g/L (0.3% dry weight (DW)) and then 4-fold concentrated by centrifugation (1.2% DW). The obtained biomass was packed in 0.5 L PET bottles and transported to the laboratories of ProdAl S.c.a.r.l. (University of Salerno, Fisciano, Italy) by courier within 24 h, under refrigerated conditions. All the experimental trials were performed within 2 days from delivery. The initial electrical conductivity of algae suspension was about 1.78 ± 0.03 mS/cm at 25 °C (Conductivity meter HI 9033, Hanna Instrument, Milan, Italy).

### 2.2. Cascade of Pulsed Electric Fields and High-Pressure Homogenization Treatments

Figure 1 schematizes the used biorefinery process for *C. vulgaris* microalgae, integrating the cascaded combination of pulsed electric fields and high-pressure homogenization technologies (PEF + HPH).

Specifically, freshly prepared microalgal biomass (0.5 L at 1.2% DW) underwent a PEF pre-treatment in a bench-scale continuous flow unit, as described in detail in our previous works [5,24,26]. Suspensions were circulated through the PEF system at a controlled flow rate of 2 L/h, employing a peristaltic pump (model PU-2080, Jasco Europe, Cremella, Italy). A stainless-steel coil (3.9 mm of inner diameter, 0.5 m of length) immersed in a water heating bath (Thermo Haake DC 10, Henco Srl, Venice, Italy), set at 25 °C, enabled controlling the inlet temperature of the suspensions. The PEF system was equipped with four co-linear treatment chambers, hydraulically connected in series and organized in two modules, with monopolar square wave pulses being delivered to microalgae suspension by a high voltage pulsed power (20 kV–100 A) generator (Diversified Technology Inc., Bedford, WA, USA). The applied voltage (0–30 kV/cm), pulse width (1–10 μs), and pulse repetition rate (1–1000 Hz) can be set independently, and only limited by the average power of 25 kW. All the experiments were carried out at fixed electric field strength (E = 20 kV/cm), total specific energy input (W_T_ = 100 kJ/kg_SUSP._), and pulse width (5 µs), which were previously identified as optimal PEF treatment conditions maximizing the release of carbohydrates and water-soluble proteins from *C. vulgaris* cells with the minimum treatment severity [5]. T-thermocouples allowed measuring product temperature at the inlet and the exit of each module of the PEF treatment zone. Under the selected electrical conditions, the maximum temperature increase of the samples at the exit of the PEF treatment system never exceeded 10 °C. Control samples of the same *C. vulgaris* suspension were obtained by pumping them through the PEF plant with the heating bath set at 25 °C, but with the PEF generator switched off.

After the PEF treatment, microalgal suspensions were collected in a 1 L flask at the exit of the PEF chamber and immediately placed in an ice-water bath for their rapid cooling down to 25 °C before undergoing the aqueous extraction process, performed according to the protocol reported in our previous work [5], which aimed to allow the diffusion of water-soluble intracellular compounds out of the cells. Samples were then centrifuged at 5700 × g for 10 min (PK121R model, ALC International, Cologno Monzese, Milan, IT) to separate the clear supernatant, representing the first output stream (S_1_), from the spent pellet. The latter was subsequently resuspended in water up to the original sample volume (0.5 L) and subjected to an ethyl acetate (EtAc) extraction step (3 h, 25 °C, 160 rpm) following the optimized protocol by Zbinden et al. [40], from which a lipid-rich phase was obtained as the second process output stream (S_2_).

The remaining pellet after the organic extraction phase was washed twice to eliminate solvent traces, and reconstituted to its initial volume with water, before being fully disrupted via HPH treatment. To this purpose, a laboratory-scale high-pressure homogenizer, described in detail elsewhere [5], was utilized. Briefly, the system consisted of a 100 µm diameter orifice valve (model WS1973, Maximator JET GmbH, Schweinfurt, Germany) through which biosuspensions were forced upon pressurization (P = 150 MPa) using an air-driven Haskel pump (model DXHF-683, EGAR S.r.l., Milano, Italy). HPH treatments were executed at a constant number of passes (n_P_ = 5), previously optimized for *C. vulgaris* to achieve complete cell disruption and, hence, full intracellular compounds release [5]. To prevent excessive heating, after each pass, the suspensions were cooled down to 25 °C in a tube-in-tube heat exchanger, located downstream of the orifice valve.

At the end of the HPH treatment, the same post-processing applied for PEF treatment was carried out, thus yielding two additional output contributions, namely an aqueous supernatant (S_3_) and an ethyl acetate extract (S_4_).

All streams collected throughout the entire biorefinery process were stored under refrigerated conditions (T = 4 °C) until further analyses. For the sake of comparison, aqueous and organic supernatants from untreated (control), individual PEF, and individual HPH treated samples were also collected and subsequently analyzed.

### 2.3. Analytical Methods

#### 2.3.1. Particle Size Distribution (PSD)

PSD of aqueous suspensions from untreated and treated (individual PEF, individual HPH, PEF + HPH) microalgae were analyzed via a MasterSizer 2000 particle size analyzer (Malvern Panalytica, Malvern, United Kingdom) at 25 °C, following a previously reported method [5,6]. In particular, the size distribution of biosuspensions was evaluated by using the Fraunhofer approximation, from which the volume-weighted mean diameter (D_4,3_) was calculated for each processing condition [41]. The parameters used in the determination of the PSD were the properties of water at 25 °C (refraction index = 1.33), which was used as a dispersant medium.

#### 2.3.2. Scanning Electron Microscopy (SEM)

The effect of different treatments on morphological characteristics of *C. vulgaris* cells was assessed by Scanning Electron Microscopy (SEM). Pellets deriving from the centrifugation of untreated and treated microalgal aqueous suspensions were prepared through a previously adopted method [5] and analyzed in a high-resolution ZEISS HD15 Scanning Electron Microscope (Zeiss, Oberkochen, Germany) at 20,000 × magnification.

#### 2.3.3. Dry Matter (DM) Content

DM of aqueous supernatants from untreated and treated microalgae was performed according to the method illustrated elsewhere [5]. Briefly, about 40 mL of each supernatant were placed in aluminum cups and dried in an oven (Heraeus, Germany) at 80 °C until a constant mass was achieved. DM was gravimetrically determined by weighing the samples before and after drying on an analytical balance (Gibertini, Italy). The dry mass content was expressed as grams of dry matter/kg of supernatant (g_DW_/kg_SUP._).

#### 2.3.4. Water-Soluble Proteins (WSP) and Carbohydrates (CH) Content of Aqueous Supernatants

The water-soluble proteins (WSP) content of aqueous supernatants (S_1_, and S_3_) from untreated and treated samples were evaluated by using the method of Lowry et al. [42], with some modifications as described in Carullo et al. [5]. Specifically, the reactive system consisted of 0.5 mL of diluted (1/2, *v*/*v* in ultrapure water) Folin–Ciocalteau reactive [43], to which 1 mL of fresh sample (supernatant), previously mixed with 5.0 mL of the reactive “C” [50 volumes of reactive “A” (2% Na_2_CO_3_ + 0.1 N NaOH) + 1 volume of reactive “B” (1/2 volume of 0.5% CuSO_4_ · 5H_2_O + 1/2 volume of 1% KNaC_4_H_4_O_6_ · 4H_2_O)] (Sigma Aldrich, Milan, Italy) were added. Absorbance was measured at 750 nm against a blank (5 mL reactive “C” + 1 mL deionized water + 0.5 mL Folin–Ciocalteau reactants), 35 min after the start of the chemical reaction, by using a V-650 Spectrophotometer (Jasco Inc., Easton, MD, United States). Bovine serum albumin (BSA) (A7030, Sigma Aldrich, Milan, Italy) was used as a protein standard. The water-soluble proteins yield (Y_WSP_) was expressed as follows (Equation (1)):Y_WSP_ (%) = (C_WSP,SUP._/C_TP,biomass_)·100(1)
where C_WSP, SUP._ is the water-soluble proteins content in the supernatant (% DW), and C_TP, biomass_ is the total protein content of *C. vulgaris* microalgae (% DW).

The total carbohydrates concentration of all the supernatants was analyzed according to the phenol-sulfuric acid method previously described by DuBois et al. [44]. Specifically, 0.2 mL of 5% (*w*/*w*) phenol and 1 mL of concentrated sulfuric acid (Sigma Aldrich, St. Louis, MO, United States) were added to 0.2 mL of diluted supernatant (Dilution Factor = 5). Samples were then incubated at 35 °C for 30 min before reading their absorbance at 490 nm against a blank of 0.2 mL 5% (*w*/*w*) phenol, 1 mL concentrated sulfuric acid, and 0.2 mL of deionized water. D-Glucose (G8270, Sigma-Aldrich, Milan, Italy) was used as a standard. The carbohydrate yield (Y_CH_) was calculated according to Equation (2):YCH (%) = (CCH,SUP./CCH,biomass)·100(2)
where C_CH, SUP._ is the carbohydrates content in the supernatant (% DW), and C_CH, biomass_ is the total carbohydrates content of *C. vulgaris* microalgae (% DW). The values of C_TP,biomass_, and C_CH,biomass_ were set equal to 61% DW and 16% DW, respectively, according to the findings from the work of Postma et al. [24].

#### 2.3.5. Lipids (LIP) Content of Organic Supernatants

Extraction and further quantification of lipids in organic supernatants (S_2_, and S_4_) from untreated and treated samples were performed according to the method illustrated by Zbinden et al. [40]. Interestingly, these authors demonstrated the potential of using ethyl acetate as the main solvent for lipids extraction from PEF-treated *A. falcatus* microalgae, in replacement of more commonly adopted hexane and isopropanol, which are more toxic and environmentally impacting [45].

For the analyses, ethyl acetate-based supernatants (50 mL) from all samples were collected in pre-weighed 100 mL round-bottomed flasks and individually evaporated to dryness under a nitrogen gas stream, by using an R-200/205 Rotavapor (BÜCHI Labortechnik AG, Flawil, Switzerland) set at 30 °C. Lipids content was gravimetrically determined from the difference in weight of samples before and after drying, and it was expressed as grams of lipids/g of dry weight biomass (g_LIP/_g_DW_ biomass). The lipids extraction yield (Y_LIP_) was calculated according to Equation (3):YLIP (%) = (CLIP,SUP./CLIP,biomass)·100(3)
where C_LIP, SUP._ is the lipids content in the organic supernatant (% DW), and C_LIP, biomass_ is the total lipids content of *C. vulgaris* microalgae (23% DW), evaluated as the complement to 100% of the protein and carbohydrates content, thus neglecting the content of minor compounds such as pigment and polyphenols. However, although this value represents a mere estimation, previous authors claimed similar lipid concentrations (26.09% DW) within *Chlorella* spp. microalgae [46], thus witnessing our choice in terms of lipids calculation basis.

#### 2.3.6. Ultraviolet-Visible (UV-Vis) Spectra Measurements

UV-Vis spectra of all aqueous and organic supernatants obtained after water and lipid extraction, respectively, were plotted as a function of the investigated range of wavelengths (λ = 200–800 nm), as previously suggested [6,36]. Prior to being analyzed, both aqueous and organic supernatants were 10-fold diluted. Characteristic peaks of proteins (λ = 290 nm for water extract, λ = 260 nm for ethyl acetate extract), yellow-to-red pigments (λ = 435 nm for water extract, λ = 430 nm for ethyl acetate extract), and chlorophyll (λ = 675 nm for water extracts; λ = 662 nm for ethyl acetate extract) were determined from spectra measurements and used to compare the effect of PEF and HPH technologies, applied individually or in cascaded combination, in terms of selectivity during either aqueous or organic extraction phases.

### 2.4. Statistical Analysis

All experiments and analyses on the extracts were carried out in triplicate, and the results were reported as mean values ± standard deviations. One-way variance (ANOVA) using Tukey’s test at a fixed significance level (*p* < 0.05) was carried out with SPSS 20 (SPSS IBM., Chicago, IL, USA) software, to assess statistically significant differences among untreated and processed samples.

## 3. Results and Discussion

This work investigated the influence of a cascade treatment on the extractability of intracellular compounds (water-soluble proteins, carbohydrates, and lipids) from microalgal suspensions of *C. vulgaris*. Cascade processing was carried out to progressively increase the level of cell damage with sequential treatments of PEF and HPH, thus eventually enhancing the overall extraction yields and selectivity of the individual extraction steps. PSD and SEM analyses of microalgal biosuspensions were used to understand the impact at the cellular level of individual treatments (PEF or HPH) or their combination in cascade. Subsequently, the release of intracellular compounds into aqueous and/or organic media was spectrophotometrically determined for the different classes of targeted compounds, together with the absorption spectra, and a systematic comparison of the different treatments, applied individually and in combination, was executed.

### 3.1. Effect of Individual PEF, Individual HPH, or Cascade Treatments on Microalgal Morphological Aspects

The volume-weighted mean diameter (D_4,3_) and SEM images of untreated and processed microalgal cells (PEF, HPH, or PEF + HPH) are reported in Figure 2 and Figure 3, respectively.

The execution of an individual PEF treatment did not lead to a statistically significant (*p* > 0.05) change in the cell mean diameter (D_4,3_ = 3.2 μm), which was only slightly reduced in comparison with the value reported by untreated microalgae (D_4,3_ = 3.45 μm). Coherently, also the particle size distribution curve did not exhibit any appreciable change (data not shown), in line with previously reported results [5], thus confirming that PEF technology can be classified as a relatively mild cell disruption method, with minimized effects on microalgal morphology. This is also evident from the comparison of the SEM images of Figure 3a,b, which show that the majority of PEF-treated microalgal cells underwent a shrinkage phenomenon, likely due to the formation of pores on the cell membrane and leakage of intracellular compounds. The increase in cell surface roughness, as well as the formation of depressions and cracks, after PEF application, was previously detected for different microalgal strains, such as *C. vulgaris* [5], *A. platensis* [6,26], *A. maxima* [47], *C. reinhardtii* [38], and *C. pyrenoidosa* [48].

Conversely, results of Figure 2 show that HPH treatment (P = 150 MPa, n_P_ = 5) induced a significant decrease (*p* ≤ 0.05) in the mean diameter D_4,3_ down to 1.95 μm, as a consequence of the mechanical fragmentation of cells and the formation of cell debris [5,16]. These observations are supported by the SEM image of Figure 3c, where the full disruption of microalgal cells and, hence, a loss in shape/structure, was detected after HPH treatment. Similar conclusions were drawn by Canelli et al. [49] in a work on the comparison between enzymatic and HPH treatments on the achieved disruption degree and the consequent release of proteins/lipids from *C. vulgaris* suspensions. Specifically, the authors reported a significant alteration in the shape of the PSD curve associated with untreated samples when subjected to HPH treatments (P = 100 MPa, n_P_ = 4), which correlated well with the observed 1.6-fold reduction in the mean particle size over control samples.

In the case of the cascaded treatments, the mean diameter lays in between the values recorded for the individual treatments by PEF and HPH (Figure 2). However, the particle size distribution curve of PEF + HPH-treated biosuspensions showed only a slight shift towards smaller sizes when compared to the untreated cells than what was induced by the individual HPH treatment, which suggests a lower cell disruption efficiency of HPH on PEF-treated cells, with the formation of larger cell debris (Figure 3d). In particular, the larger mean size of the formed debris after the cascaded treatments as compared to that observed in the case of HPH treatment alone might facilitate, presumably, the separation phase in downstream processing. Additionally, in the SEM picture of cascade-treated biomass, it is worth noting the action of organic solvent (yellow arrows), due to the induced cell wall/membrane system erosion upon lipid removal, which occurred before the final cell disintegration by HPH in the proposed biorefinery, suggesting that also lipid extraction might have affected the efficiency of the HPH disruption process.

The decrease in cell disruption efficiency of HPH in the cascaded treatments in comparison with HPH alone might, therefore, be ascribed to the stress induced in the microalgal cells by the precedent treatments, and in particular PEF application and the contact with the organic solvent during the lipid extraction phase, which might have affected the cell structure. In particular, the release of intracellular compounds due to PEF treatment, causing the cell shrinkage observed in Figure 3b, might have increased the capability of the cell deformability, hence making the cell walls more resistant to the intense fluid dynamic fields generated in the HPH process. In accordance with our results, Alvarez & Heinz [50] found that the application of PEF pre-treatment for the inactivation of *Salmonella* Senftenberg 775W and *Listeria monocytogenes* increased the resistance to cell disruption when subsequently treated by ultrasounds (US), thus reducing the efficacy of the combined approach with respect to individual US treatments.

Another possible explanation of the reduced effect of HPH in the proposed cascade approach is the formation of cell clusters, which may have occurred during the resuspension in the water of the pellet immediately after the organic extraction. However, the formation of such cell aggregates was not confirmed by SEM analysis (Figure 3d), thus suggesting that the hypothesis of microalgae structural changes upon either PEF or organic solvent contact is more plausible. However, this aspect needs to be further investigated to be fully elucidated.

### 3.2. Effect of Individual PEF, Individual HPH, or Cascade Treatment on the Recovery of Intracellular Compounds

#### 3.2.1. Extraction Yields

The dry matter content (DM) of the aqueous supernatants from untreated and treated (individual treatments and combination PEF + HPH) biomass is depicted in Figure 4, with the insert showing the appearance of the supernatants obtained in the different cases.

Remarkably, the supernatant from electrically treated samples was as clear as the one obtained from intact cells (control sample), thus corroborating the inability of PEF to cause the leakage of water-insoluble pigments from microalgae during aqueous extraction, as previously demonstrated by Carullo et al. [6] and Grimi et al. [36].

The release of intracellular compounds significantly (*p* ≤ 0.05) increased with the intensity of the applied treatment, with a maximum value detected after HPH processing (DM = 7.66 g_DM_/kg_SUP._), which was approximately 2.8 times higher than that observed for PEF-treated samples. As previously observed [5], it is likely that the finest fraction of the cell debris produced upon full mechanical cell disintegration remained suspended in the supernatant after centrifugation, thus contributing to an overestimation of the real amount of dry matter content and the opacity of the sample, as testified by the dark green coloration of the supernatant (Figure 4). Moreover, the complete cell disruption may have caused the release of different classes of intracellular compounds, including chlorophyll, thus making the HPH treatment alone an unsuitable method for the selective extraction of valuable molecules from microalgal biosuspensions.

The application of HPH in the cascade scheme (PEF + HPH) led to a small but significant (*p* ≤ 0.05) increase in the dry matter content of supernatant with respect to PEF-treated samples, because of the increased release of water-soluble compounds into the external medium. Correspondingly, the supernatant obtained from the combined treatment showed a light green coloration (Figure 4), which may be attributed to the solubilization of a smaller amount of cell debris as compared to HPH treatment alone, thus reinforcing the previously postulated hypothesis that solid–liquid separation becomes easier when applying the investigated technologies in a cascade mode.

The release of intracellular compounds after either aqueous (mainly water-soluble proteins, WSP, and carbohydrates, CH) and organic (mainly lipids, LIP) extraction was quantified, with the results reported in Figure 5.

The leakage of intracellular compounds from untreated cells was extremely low during water extraction, being driven only by spontaneous cell lysis or concentration gradient between cells and external medium [5,51]. Interestingly, the greater capability of ethyl lactate to penetrate undamaged *C. vulgaris* cellular structure enabled significantly higher extraction yields of lipids (Figure 5c) than the yields of WSP and CH detected in the case of water extraction (Figure 5a,b).

Figure 5a also shows that PEF technology resulted as scarcely efficient in extracting proteins, especially when compared to more intense treatments, such as bead milling or HPH, coherently with the results reported here for HPH treatment and previously observed for the mechanical permeabilization of hard-structured microalgal strains [24,31]. Hence, it can be hypothesized that the pores formed on the microalgal cell membrane during PEF treatment are not sufficiently large to allow the release of high-molecular-weight proteins, which may likely have remained trapped inside the cell, or bounded to the cell wall, which was not affected by the electrical treatment [5,52,53].

The results of Figure 5a are consistent with the findings of Grimi et al. [36], who investigated the application of PEF in a sequential cascade mode with other disruption techniques, such as high voltage electrical discharges (HVED), ultrasounds (US), and high-pressure homogenization (HPH). The authors found that the latter was the most efficient disruption method leading to the highest amount of extracted proteins (91%), while PEF only showed an efficiency of 5%, a value which was greater than the supplementary contributions of HVED and US.

It is also worth noting from Figure 5a that the amount of water-soluble proteins extracted by the cascade approach was significantly (*p* ≤ 0.05) smaller than that obtained by the HPH treatment alone, despite inducing a 6-fold increase with respect to PEF treatment alone. This behavior could be attributed to the contact between proteins and ethyl acetate during the lipid extraction phase, which likely triggered proteins degradation/denaturation phenomena, making insoluble a significant fraction of the soluble proteins, with a subsequent loss of their techno-functional properties [13]. Nevertheless, further investigations are strongly recommended in order to confirm this hypothesis.

Smaller molecules like carbohydrates seem to easily move across the pores formed by PEF treatment during aqueous extraction (Figure 5b), hence leading to a significant (*p* ≤ 0.05) increase in the extraction yield (Y_CH, PEF_ = 24.3%) over untreated samples. Previous studies demonstrated that PEF was capable of unlocking low molecular weight compounds (e.g., carbohydrates) during water diffusion at yields comparable to those arising from highly disruptive techniques [5,24]. Eventually, the greater extraction efficiency of PEF towards carbohydrates rather than water-soluble proteins might help to explain the trend observed in Figure 5b, since no statistical differences were detected between yields from individual HPH treatment and the cascade approach (*p* > 0.05). In line with our previous work, dealing with the combined effect of PEF and high-shear homogenization on the extractability of valuable compounds from *A. platensis* [6], similarities in terms of WSP and CH extraction yields (Y_WSP, HPH_ = 54.01%, Y_CH, HPH_ = 49.05%) achieved after mechanical fragmentation of microalgae are likely due to the undifferentiated release of intracellular matter from cells, thus negatively impacting the selectivity of HPH processing alone within a microalgal biorefinery.

Figure 5c shows the yield of recovered lipids from the untreated samples, and those subjected to PEF alone, HPH alone, and cascade treatments. The electroporation of cell membranes caused a significant (*p* ≤ 0.05) increase in the extraction yield of lipids in comparison with controls, in agreement with previously published papers reporting that the enhancement of lipids extraction by PEF is effective [10,20,40,48,54]. For instance, in the work of Lai et al. [54], the application of a PEF treatment of constant intensity (30.6 kWh/m^3^) on *Scenedesmus* biosuspensions yielded 3.1-fold more crude lipids and fatty acid methyl esters (FAME) than those from conventional solvent extraction.

Coherently with the results of Figure 2 and Figure 3 highlighting the greater disruption efficiency of the mechanical process as compared to the electrical one, further enhancements in lipids recovery were detected when HPH treatments alone were performed, with values of extraction yields reaching around 70%.

Interestingly, when PEF and HPH were applied in the cascade mode, thus involving a double stage of solvent extraction, the maximum achievable Y_LIP_ was detected (100%). However, it must be taken into account that in the repeated extraction procedure, the organic solvent may have become enriched also with a certain amount of water-insoluble proteins, which naturally separated from the aqueous supernatant, together with the pellet, after the centrifugation of the biosuspension following water extraction. Therefore, in future works, it is recommended that more sophisticated methods of lipids determination (e.g., gas chromatography–mass spectrometry analysis) within the organic extracts are executed to better assess the effectiveness of alternative cell disruption methods, such as PEF and HPH, on the extraction yield of single lipidic compounds from microalgae.

#### 3.2.2. UV-Vis Absorption Spectra

Results from UV-Vis analyses are reported in Figure 6, for both aqueous (a) and organic supernatants (b), respectively, recovered from untreated and treated samples.

While the aqueous extracts from the fresh microalgal suspensions exhibited a completely flat profile, the PEF-treated samples exhibited a single peak at around 290 nm, corresponding to the extraction of a small amount of water-soluble proteins. This confirms the complete absence of pigments extraction in PEF-treated samples, thus evidencing a selective behavior towards specific intracellular compounds, but lower yields than more destructive methods, such as HPH (Figure 6a). The HPH treatment alone, instead, led to an instantaneous release of all the intracellular compounds, with absorption spectra showing the presence not only of proteins but also of yellow-to-red pigments (435 nm) and chlorophyll (675 nm), as previously reported [36]. In the case of cascaded treatments, a lower intensity of the pigments peak was observed in the aqueous extracts, likely due to their selective removal by the organic solvent during the intermediate lipid extraction phase. In general, the spectra measurements reported in Figure 6a are in good agreement with the supernatant images reported in the insert of Figure 4.

The spectra of the organic phases reported in Figure 6b clearly show how the complete cell permeabilization induced by HPH treatment also caused the highest extractability of all the pigments. Moreover, almost overlapped spectra were observed in the case of untreated and PEF-treated samples. As in the case of Figure 6a, the amount of pigments solubilized in the organic solvent by the cascaded treatments resulted in being lower than that achieved for fully ruptured cells by HPH alone, thus supporting the hypothesis of increased selectivity for the cascade biorefinery scheme imparted by the integration of the PEF treatment. However, it is worth mentioning that all the ethyl acetate-based extracts of Figure 6b were characterized by a small peak at 260 nm, which is likely induced by the extraction of a protein fraction, which confirms the hypothesis of proteins interference throughout gravimetric measurements for lipids determination.

## 4. Conclusions

The results reported in this study have shown the potentiality of the integration of PEF technology in a microalgal biorefinery scheme, in cascade operation with HPH (PEF + HPH), which enabled not only an efficient extraction of valuable molecules from microalgae *C. vulgaris* but also the selective separation of distinct classes of compounds throughout the planned sequence of processing steps. Coupling PEF and HPH technologies enabled reaching extraction yields of carbohydrates and lipids comparable or even higher than those observed when fully disrupting microalgal cells by HPH treatments alone, together with a higher purity of the obtained extracts. Moreover, the formation of cell debris from the cascade approach is significantly lower than through HPH alone, thus potentially leading to an economical benefit during separation phases in downstream processing.

However, additional studies are required to better elucidate the role of investigated technologies, applied in cascade mode, on the proteins extraction phase, being the most critical step of *C. vulgaris* biorefinery, as well as to properly tune the biomass concentration in treated suspensions to optimize process energy consumptions. Furthermore, the possibility to extend the same cascade approach to other microalgae species should be also investigated.

## Figures and Tables

**Figure 1 foods-11-00471-f001:**
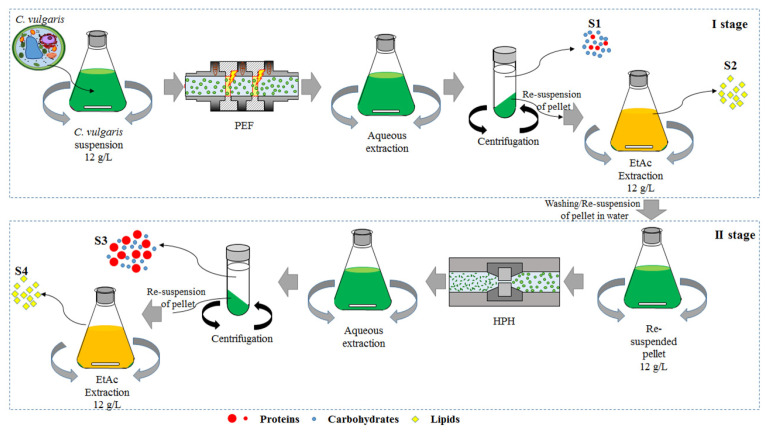
Schematic representation of the “cascade biorefinery” of *C. vulgaris* microalgae proposed in this study. I stage: PEF-assisted extraction; II stage: HPH-assisted extraction. EtAc: ethyl acetate.

**Figure 2 foods-11-00471-f002:**
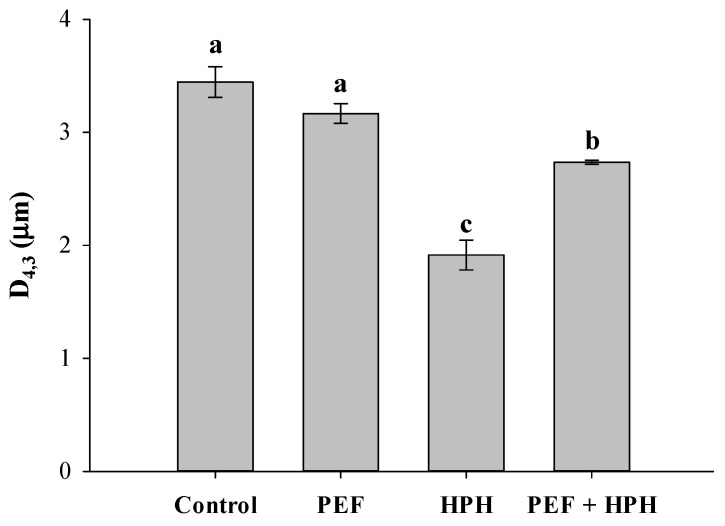
Volume-weighted mean diameter (D_4,3_) of *C. vulgaris* suspensions untreated (control) and treated by PEF alone, HPH alone, and combined PEF + HPH. Different letters above the bars indicate statistically significant differences among the samples (*p* ≤ 0.05).

**Figure 3 foods-11-00471-f003:**
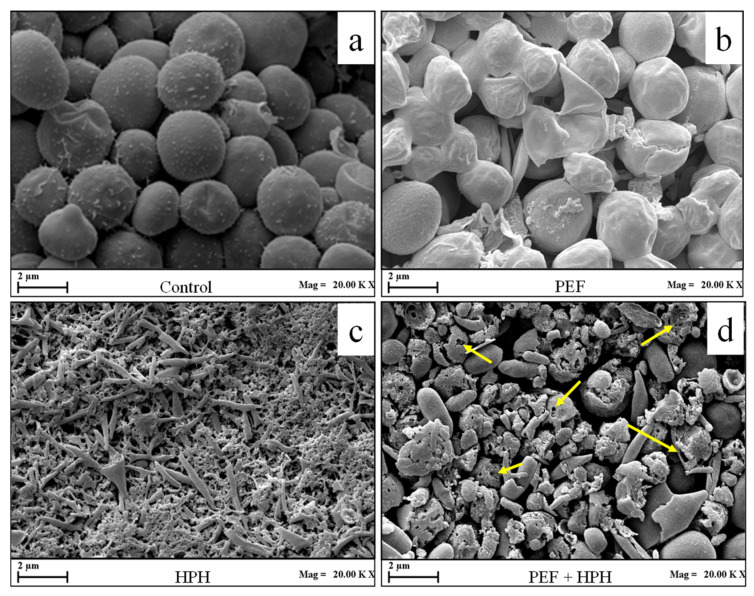
Scanning electron microscopy (SEM) of *C. vulgaris* cells, before (**a**) and after the application of PEF alone (**b**), HPH alone (**c**), and combined PEF + HPH (**d**).

**Figure 4 foods-11-00471-f004:**
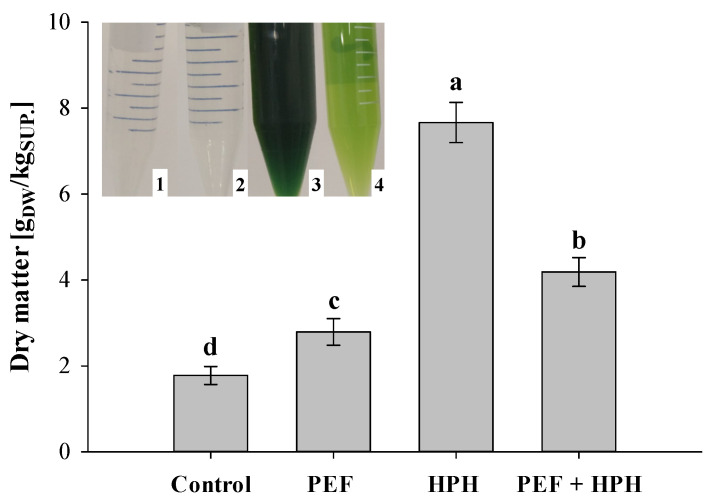
Dry matter content of aqueous supernatants from *C. vulgaris* suspensions untreated and treated by PEF alone, HPH alone, and with the combination of PEF + HPH. Different letters above the bars indicate significant differences among the samples (*p* ≤ 0.05). Insert shows the pictures of aqueous supernatants obtained after centrifugation of microalgae untreated (1) and treated by PEF alone (2), HPH alone (3), and PEF + HPH (4).

**Figure 5 foods-11-00471-f005:**
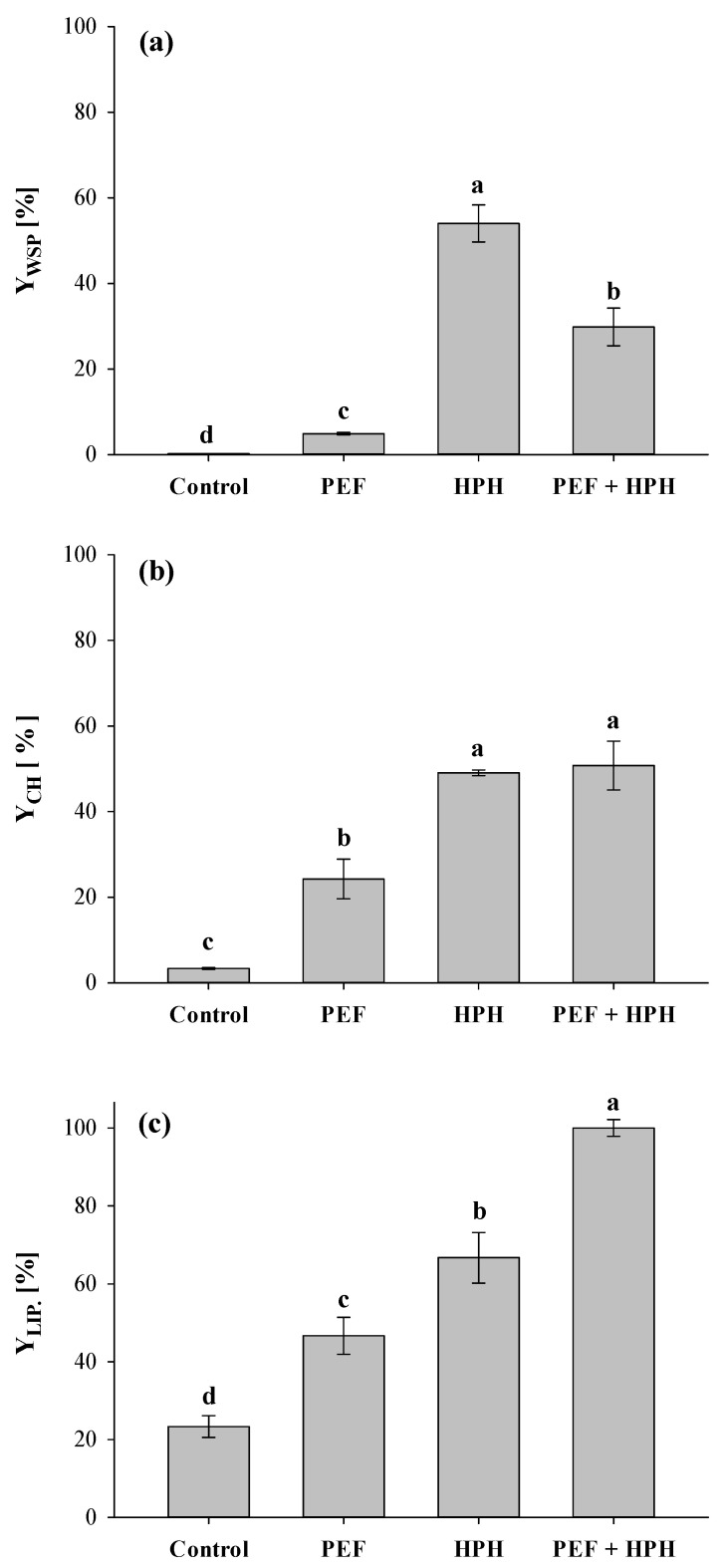
Yields of water-soluble proteins (**a**), carbohydrates (**b**), and lipids (**c**) extracted from *C. vulgaris* suspensions untreated, and treated by PEF alone, HPH alone, and combined PEF + HPH. Different letters above the bars indicate significant differences among the samples (*p* ≤ 0.05).

**Figure 6 foods-11-00471-f006:**
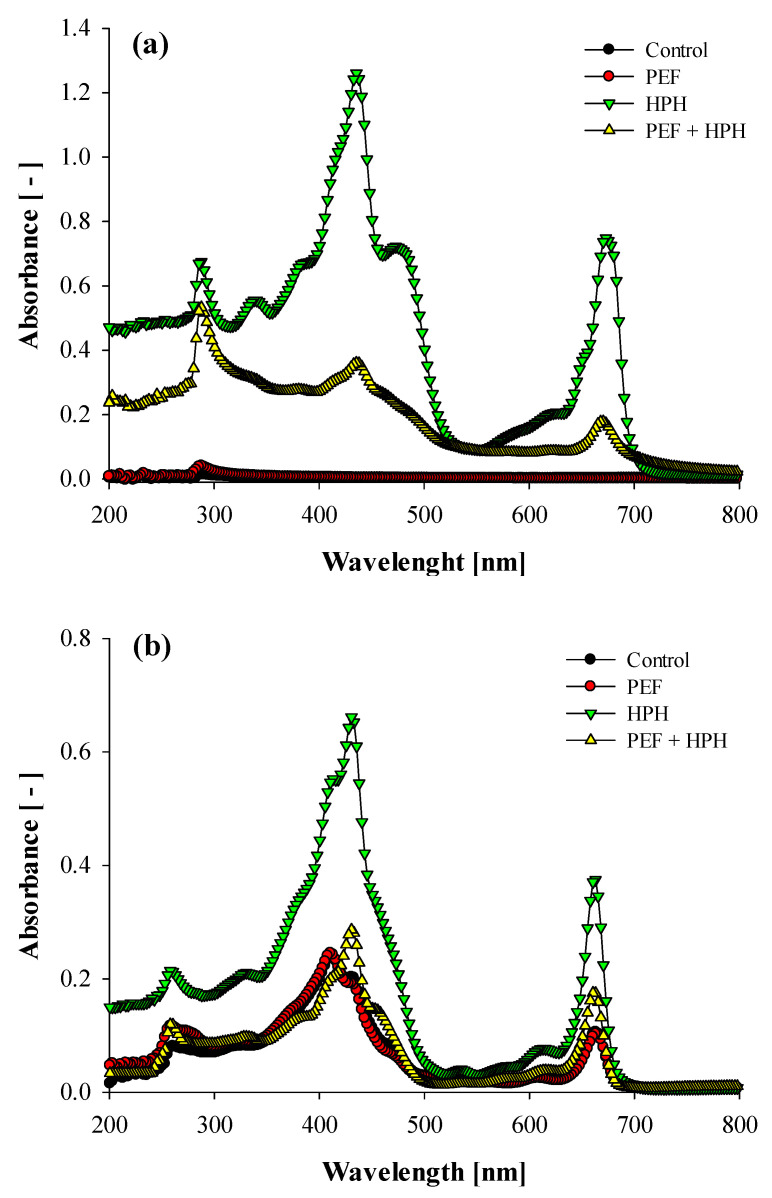
UV-Vis absorption spectra of aqueous (**a**) and organic (**b**) supernatants from *C. vulgaris* suspensions untreated (control), and treated by PEF alone, HPH alone, and combined PEF + HPH.

## Data Availability

Data are contained within the article.

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
