# Peer review of "Application of Pulsed Electric Fields and High-Pressure Homogenization in Biorefinery Cascade of C. vulgaris Microalgae"

_foods, 2022, doi:10.3390/foods11030471_

Round 1

Reviewer 1 Report

I have reviewed ‘Application of pulsed electric fields and high-pressure homogenization in biorefinery cascade of C.vulgaris microalgae’. In this paper, the author applied a cascading processing methods (pulsed electric fields and high-pressure homogenization) to aid protein, carbohydrates, and lipid extraction from microalgae and get pretty promising results. Overall, the paper is well written with detailed description of methods, results, and discussion, and the figures are well drafted and appealing. In the meantime, it is a pretty interesting paper and provides some valuable information to the microalgae industry. Some minor comments:

  1. Figure 5: the highest mean value should be marked as ‘a’ as the significant level, etc.
  2. Line 197: what modifications have been made and why making these modifications?
  3. In the experimental section, please add more description of what the control means and how was conducted.
  4. The author provided the yield of protein, carbohydrates, and lipids after biorefinery treatment, how about the purity of each product?
  5. How did the author’s processing methods compare with other processing methods such as ultrasound or microwave aided extraction?
  6. What is the possible application of pulsed electric fields and high-pressure homogenization for microalgae value compounds extraction in the real industry, any challenges?

Author Response

Response to Reviewer 1:

I have reviewed ‘Application of pulsed electric fields and high-pressure homogenization in biorefinery cascade of C.vulgaris microalgae’. In this paper, the author applied a cascading processing method (pulsed electric fields and high-pressure homogenization) to aid protein, carbohydrates, and lipid extraction from microalgae and get pretty promising results. Overall, the paper is well written with detailed description of methods, results, and discussion, and the figures are well drafted and appealing. In the meantime, it is a pretty interesting paper and provides some valuable information to the microalgae industry. Some minor comments:

Figure 5: the highest mean value should be marked as ‘a’ as the significant level, etc.

Now the highest mean values appearinmg in Fig. 4 and 5 have been marked with the letter “a”.

Line 197: what modifications have been made and why making these modifications?

Modifications regarded only the set reaction time. This change was driven by previous spectra measurements of the whole reaction mixture, which demonstrated that the greatest color intensity developed upon the reaction between Folin-Ciocalteau reactants and water-soluble proteins was achieved after 35 min from the mixing phase, instead of 30 min indicated in the work of Lowry et al. [Ref. Lowry, O.H.; Rosebrough, N.J.; Farr, A.L.; Randall, R.J. Protein measurement with the Folin phenol reagent. J. Biol. Chem. 1951, 193, 265-275].

In the experimental section, please add more description of what the control means and how was conducted.

We thanks the Reviwer for this comment since it is true that, in the previous version of the manuscript, no information was given at all about how control samples were obtained. Now, this information has been added to section paragraph 2.2 in order to clarify this aspect.

The author provided the yield of protein, carbohydrates, and lipids after biorefinery treatment, how about the purity of each product?

We agree with the Reviewer that, in this work, the purity values of each achieved extract were not calculated. However, the authors would like to point out that the absorption spectra of Figure 6 could address a qualitative discussion about the effects of PEF and HPH treatments, applied alone or in their combination, on the quality and purity of outlet streams collected throughout the whole biorefinery process of C. vulgaris microalgae.

How did the author’s processing methods compare with other processing methods such as ultrasound or microwave aided extraction?

At this stage, a comparison between PEF and HPH technologies, applied in the current work, and some other disruptive methods, such as ultrasound (US) and microwave (MW) treatments, in terms of extraction efficiency of intracellular compounds from C. vulgaris microalgae, was out of the scope of this work, thus avoiding any speculation on the matter. Nevertheless, it must be inferred that an accurate comparison with other processing methods would require the adoption of at least similar extraction protocols, as well as an economic/energetic analysis.

What is the possible application of pulsed electric fields and high-pressure homogenization for microalgae value compounds extraction in the real industry, any challenges?

Both technologies can be used in contours flow and scaled up. However, the challenge now could be represented by the initial cost investment for PEF and HPH technologies. Moreover, further research is needed in order to determine optimal processing conditions for the different type of microalgae species as well as the optimal biomass concentration in order to reduce the processing costs.

Reviewer 2 Report

I think the work is very interesting but some small changes are necessary to improve it.

Section INTRODUCTION

The introduction is interesting but it need to introduce more appropriate  references:

Line 36 Matos, Â.P. The Impact of Microalgae in Food Science and Technology. J Am Oil Chem Soc 94, 1333–1350 (2017). https://doi.org/10.1007/s11746-017-3050-7

Rahman K.M. (2020) Food and High Value Products from Microalgae: Market Opportunities and Challenges. In: Alam M., Xu JL., Wang Z. (eds) Microalgae Biotechnology for Food, Health and High Value Products. Springer, Singapore. https://doi.org/10.1007/978-981-15-0169-2_1

Line 38 Carbone, D.A.; Pellone, P.; Lubritto, C.; Ciniglia, C. Evaluation of Microalgae Antiviral Activity and Their Bioactive Compounds. Antibiotics 202110, 746. https://doi.org/10.3390/antibiotics10060746

de Jesus Raposo MF, Costa de Morais RM, de Morais A, Health applications of bioactive compounds from marine microalgae,Life Sciences,Volume 93, Issue 15, 2013,Pages 479-486,

ISSN 0024-3205,https://doi.org/10.1016/j.lfs.2013.08.002.

Section MATERIAL AND METHODS

  • In paragraph 2.1 you should add the growth light intensities of Chlorella it is an important parameter that can influence the estraction
  • In paragraph 2.3.2 line 184 you talk about a little bit about the method
  • Paragraph 2.3.4, you can modified the yield formula and add *100.

E.G. YCH(%)= (Cchsup/Cchbiomass)*100

  • You repeated twice paragraph 2.3.5 (line 224 and 246)
  •  

Section RESULT AND DISCUSSION

  • I think the results are very interesting and figures are very clear but the problem is that the text is too complicated to read. I suggest to write some information (for example about the figures) in table and not in the text.

Section CONCLUSION

I suggest to add some ideas about the prospective utilization and future works about these technique

Author Response

Response to Reviewer 2:

I think the work is very interesting but some small changes are necessary to improve it.

Section INTRODUCTION

The introduction is interesting but it need to introduce more appropriate  references:

  • Line 36 Matos, Â.P. The Impact of Microalgae in Food Science and Technology. J Am Oil Chem Soc 94, 1333–1350 (2017). https://doi.org/10.1007/s11746-017-3050-7
  • Rahman K.M. (2020) Food and High Value Products from Microalgae: Market Opportunities and Challenges. In: Alam M., Xu JL., Wang Z. (eds) Microalgae Biotechnology for Food, Health and High Value Products. Springer, Singapore. https://doi.org/10.1007/978-981-15-0169-2_1
  • Line 38 Carbone, D.A.; Pellone, P.; Lubritto, C.; Ciniglia, C. Evaluation of Microalgae Antiviral Activity and Their Bioactive Compounds. Antibiotics 2021, 10, 746. https://doi.org/10.3390/antibiotics10060746
  • de Jesus Raposo MF, Costa de Morais RM, de Morais A, Health applications of bioactive compounds from marine microalgae,Life Sciences,Volume 93, Issue 15, 2013,Pages 479-486, ISSN 0024-3205,https://doi.org/10.1016/j.lfs.2013.08.002.

We kindly appreciate the suggestion given by the Reviewer to improve and update the Introduction section with some striking research were cited in the text and included within the list of references.

Section MATERIAL AND METHODS

In paragraph 2.1 you should add the growth light intensities of Chlorella it is an important parameter that can influence the estraction.

According to the Reviewer’s comment, the light intensity adopted during C. vulgaris cultivation has been added.

In paragraph 2.3.2 line 184 you talk about a little bit about the method

The authors decided not to include much information about the method followed for preparing samples for SEM analyses, since in one of our previous works cited in the text [Carullo, D.; Abera, B.D.; Casazza, A.A.; Donsì, F.; Perego, P.; Ferrari, G.; Pataro, G. Effect of Pulsed Electric Fields and High- Pressure Homogenization on the aqueous extraction of intracellular compounds from the microalgae Chlorella vulgaris. Algal Res. 2018, 31, 60–69, doi: 10.1016/j.algal.2018.01.017] we provided all the details in case one would like to reproduce our results.

Paragraph 2.3.4, you can modified the yield formula and add *100.

E.G. YCH(%)= (Cchsup/Cchbiomass)*100

The formulas have modified accordingly.

You repeated twice paragraph 2.3.5 (line 224 and 246).

Now, in the new version of the manuscript, the paragraph “UV-Vis spectra measurements” was referred to as 2.3.6.

Section RESULT AND DISCUSSION

I think the results are very interesting and figures are very clear but the problem is that the text is too complicated to read. I suggest to write some information (for example about the figures) in table and not in the text.

In the opinion of the authors, the use of tables to be filled in with some information (as gently requested by the Reviewer), would unavoidably lead to the repetition of data already reported in previous Figures, which is something that must be avoided in a paper.

Section CONCLUSION

I suggest to add some ideas about the prospective utilization and future works about these technique

Now the text has been revised accordingly.